# Amoxicillin/Clavulanic Acid in Transrectal Biopsy of the Prostate—An Alternative in Times of Ciprofloxacin Obsolescence and Fosfomycin Limitation?

**DOI:** 10.3390/antibiotics13100940

**Published:** 2024-10-06

**Authors:** Philipp J. Spachmann, Sophie E. Fischer, Christopher Goßler, Stefan Denzinger, Maximilian Burger, Johannes Breyer, Wolfgang Otto, Marco J. Schnabel, Johannes Bründl, Bernd Rosenhammer

**Affiliations:** 1Department of Urology, Caritas St. Josef Medical Center Regensburg, University of Regensburg, 93053 Regensburg, Germany; 2Urologie Landau Isar and Urologie Plattling, 94405 Landau/Isar, 94447 Plattling, Germany

**Keywords:** amoxicillin/clavulanic acid, antibiotic prophylaxis, infectious complications, transrectal prostate biopsy

## Abstract

Recently, the German Federal Institute for Medicines and Medical Products restricted the use of fosfomycin in transrectal biopsy of the prostate (TRBP). Accordingly, the need for other antibiotic agents for prophylaxis in TRBP is urgent since antibiotic prophylaxis is mandatory in accordance with these guidelines. After the restriction of the use of ciprofloxacin, and before the use of fosfomycin in Germany was falsely allowed, amoxicillin/clavulanic acid was evaluated as an alternative for antibiotic prophylaxis in TRBP. Regarding hospitalization for post-interventional infections, 359 patients at the Department of Urology of the University of Regensburg, at Caritas-St. Josef Medical Center as a single center, underwent TRBP between 2 July 2019 and 30 June 2020. Regarding antibiotic prophylaxis, the post-interventional hospitalization rate due to bacterial complications was relevant. Of the 359 patients, 10 (2.8%) had an infection requiring hospitalization post-TRBP. A total of 349 (97.2%) patients had no infection-related hospitalization. This corresponds to an incidence rate of only 2.8%. Referring to the previous infection rates under the now obsolete ciprofloxacin, amoxicillin/clavulanic acid can show a similar, if not tendentially even lower, risk of infection, and so this substance can be an alternative for antibiotic prophylaxis in TRBP. Another advantage is that, according to the WHO’s AWaRe classification, amoxicillin/clavulanic acid is one of the so-called Access antibiotics. This study is limited as rectal swabs and urine cultures were not performed on every patient before TRBP.

## 1. Introduction

In 2019, the European Medicines Agency (EMA) issued warnings against the use of ciprofloxacin, including as antibiotic prophylaxis in transrectal biopsy of the prostate (TRBP), and a restriction was published in Germany in April 2019 [1].

Recently, the PREVENT trial showed, in contrast to current guideline recommendations, that there was no significant difference regarding infectious complications between TRBP and the perineal approach. In the PREVENT trial, infectious complications of transperineal biopsies of the prostate without antibiotic prophylaxis were compared with those of transrectal biopsies with targeted prophylaxis. In a randomized total cohort of 658 patients, 0 patients in the transperineal group and 4 in the transrectal group (1.4%) showed infections after a biopsy of the prostate (*p* = 0.059) [2]. This underlines the persisting relevance of TRBP and the need for adequate antibiotic prophylaxis.

As there were no suggestions for evidence-based alternative antibiotic prophylaxis, and fosfomycin was not yet—as now known, falsely, in Germany [3]—recommended, urologists faced the challenge of independently defining a new prophylactic substance, as prostatitis after TRBP might be the most feared complication of this procedure. Due to this, the guidelines recommend performing TRBP only under antibiotic prophylaxis [4,5]. Also, surveillance for infectious complications is of eminent relevance in the search for new options for prophylaxis. 

An important consideration was to choose a substance that can be applied orally. This was due to in-house processes as most patients were sent to the hospital for the first time, and because of the high number of patients to be biopsied, the in-house structure allowed no pre-intervention intravenous application of prophylaxis. Since oral cephalosporines have a low oral bioavailability, the enterococci gap exists and the local resistance rates against cotrimoxazole were higher than 20%, the use of this substance could not be propagated properly.

In accordance with the principles of antibiotic stewardship, it was necessary to choose a substance that, on the one hand, is approved for infections of the urinary tract, such as prostatitis, and, on the other hand, is appropriate for the local level of resistance. This is to prevent the further increase in resistance rates to antibiotics or antibacterial drugs and to prevent increased rates of multiresistant bacteria. For Bavaria as a whole, *E. coli* in offices was the most common cause of urinary tract infections, and there was a resistance level to amoxicillin/clavulanic acid of 13.9% in 2019, and in 2021, after this study was conducted, it was 9.8% regionally for the Oberpfalz district [6,7].

At the same time, the calculated use of antibiotics in particular must be adapted to the overall development of resistance in accordance with the principles of antibiotic stewardship. For this purpose, the World Health Organization (WHO) introduced the AWaRe classification for antibiotics, which divides antibiotics into the following groups: “Access”, for generally recommended use, if possible, as they are effective against most bacteria and are generally available; “Watch” for use of the substance while monitoring the resistance situation, as these antibiotics are of great relevance for the treatment of infections but have, at the same time, higher resistance induction potential; and “Reserve” for the group of antibiotics that are used to treat multiresistant bacteria in particular, and should, therefore, only be used very restrictively with simultaneous monitoring of their use [8].

The aim of the present study was to investigate infectious hospitalization complications after TRBP with amoxicillin/clavulanic acid at a single center in a university.

## 2. Results

The mean age of the entire collective was 66.94 years [SD: 8.0]; 137 patients (38.2%) in the collective were 70 years of age or older and could be defined as geriatric. In total, 112 (31.2%) patients had a re-biopsy: 82 (22.8%) had a second biopsy, 19 (5.3%) had a third, six (1.7%) had a fourth, three (0.8%) had a sixth and two (0.6%) had a seventh biopsy. The mean interval from the last biopsy was nine months [SD:21.3].

Most patients, 303 (84.4%) in total, received an MRI fusion biopsy; the median number of biopsies carried out was 14, and the remaining 56 (15.6%) patients received a random biopsy with 12 biopsies taken.

The mean prostate volume was 55.37 mL [SD: 29.2], ranging between 13 and 241 mL. In total, 310 patients (86.35%) had benign prostatic hyperplasia (volume > 30 cm^3^), and the prostate-specific antigen PSA had a minimum of 0.25 ng/mL and a maximum of 2351 ng/mL with a mean value of 7.8 ng/mL. A total of 165 (46%) patients showed an inflammatory infiltrate in the histology of TRBP. For other patient characteristics, see Table 1.

Of the 359 patients, 10 (2.8%) had a post-interventional infection requiring hospitalization. The remaining 349 (97.2%) patients had no complications after TRBP.

Patients with infections were on average 65.4 years old [SD:7.77], and there were no significant differences between patients with and without infection after a prostate biopsy according to age at TRBP (*p* = 0.624). In the cohort of geriatric patients with an age of >70 years, three patients had an infectious hospitalization, also without any significant differences comparing geriatric and non-geriatric patients (*p* = 0.428).

An increasing number of TRBPs themselves (*p* = 0.955) or the time interval from the last biopsy (*p* = 0.065) did not result in a significantly higher risk of an infection.

All patients with infectious complications received an MRI fusion biopsy, but no significant correlation between an increase in the number of biopsies and the occurrence of an infection could be observed (*p* = 0.568).

Neither the median prostate volume (*p* = 0.142) nor the median PSA value (*p* = 0.332) showed statistically significant differences between the two groups; also, inflammatory infiltrates in histology, meaning chronic prostatitis, did not result in significant differences regarding infection (*p* = 0.086).

Also, neither the detection of prostate cancer (*p* = 0.086), nor a situation of immunosuppression (*p* = 0.919), cardiovascular disease (*p* = 0.179), antibiotic treatment within six months before TRBP (*p* = 0.819), urinary tract infection within 12 months before TRBP (*p* = 0.819, urogenital risk factors (*p* = 0.709) or a suspicious finding in the DRE (*p* = 0.212) showed statistically significant differences in the risk of infectious hospitalization according to TRBP. Likewise, there were no statistically significant differences for the group in which a rectal swab was performed before TRBP (*p* = 0.204), as seen in Table 1.

A significantly increased risk of infection after prostate biopsy could be demonstrated for diabetics as a predisposing illness for a post-interventional infection (*p* = 0.032).

Nine of the ten cases with an infection requiring hospitalization had a positive urine culture, and in every case of a positive culture, *E. coli* was found.

The antibiogram of urine cultures regarding the sensitivity to the prophylaxis amoxicillin/clavulanic acid showed five of the infected cases without sensitivity against amoxicillin/clavulanic acid and four cases with sensitivity against amoxicillin/clavulanic acid in high doses, and of that, one case of *E. coli* was a 3MRGN; this means that there was resistance to aminopenicillins, cephalosporins of the third and fourth generation and to quinolones. In one event there was no finding of bacteria.

In nine cases, blood cultures were sampled, and seven cases were positive (77.8%), with four *E. coli* cases being resistant to amoxicillin/clavulanic acid, and three being sensitive.

## 3. Discussion

Ten patients required hospitalization for infectious complications following antibiotic prophylaxis with amoxicillin/clavulanic acid, resulting in a rate of just 2.8 percent, which seems to be within or even below the previously described hospitalization rates under prophylaxis with ciprofloxacin [9,10,11,12,13]. However, the study’s limitations must be noted; firstly, the number of cases of infectious complications was very low, which, secondly, means that a multivariate analysis was not statistically possible. A trend can therefore be seen, but there is no statistically adequate comparability due to the statistical limitations of the present study.

Also, the TRBP time was not recorded, and which TRBP operator performed which and how many TRBPs was not recorded, although all of them were experienced examiners. Regarding the configuration of the prostate, the above-mentioned data on size and the results of the DRE were available, but data were not available for other anatomical abnormalities of the rectal access or the symmetry or asymmetry of the prostate. It is important to note that this study is limited by its single-center character and lacks comparison between antibiotic substances. Another limitation is that rectal swabs and urine cultures were not performed on every patient before TRBP.

According to the principles of antibiotic stewardship and the WHO’s AwaRe classification, amoxicillin/clavulanic acid is, therefore, adequate for antibiotic prophylaxis, the choice of which can also prevent the further spread of resistance to, for example, intravenously administered substances of the third and fourth generation of cephalosporins or, e.g., piperacillin/tazobactam. Interestingly, oral fosfomycin–trometamol is part of the “Watch” group, so the use of this substance, which is often used worldwide for prophylaxis in TRBP, should be more restrictive than amoxicillin/clavulanic acid, which in turn speaks in favor of the use of amoxicillin/clavulanic acid, as it is part of the “Access“ group [8].

However, a negative influence on the microbiome must be assumed, especially resulting from a background of the prolonged administration of antibiotic prophylaxis.

Interestingly, in the observed collective, there was no significant association between the probability of a prostatitis after antibiotic prophylaxis with amoxicillin/clavulanic acid for biopsy regarding age (*p* = 0.624), geriatric patients at an age of more than 70 years (*p* = 0.428), the number of TRBPs themselves, or the number of biopsies being taken (number of TRBP itself *p* = 0.955; number of biopsies *p* = 0.568).

Four of nine patients presented with an infection requiring hospitalization after TRBP and a positive urine culture showing *E. coli* with sensitivity against amoxicillin/clavulanic acid, which is remarkable. Perhaps the duration of the application of amoxicillin/clavulanic acid was not long enough, or perhaps the medication was not taken as recommended.

On the other hand, this might show that, even with adequate prophylaxis, an infection cannot be completely ruled out by performing TRBP.

A high correlation could be observed in patients with diabetes as a comorbidity, as three patients who were hospitalized due to an infection were diabetic (*p* = 0.032).

A significantly higher potential risk for developing an infection requiring hospitalization after transrectal biopsy could be observed in patients with diabetes. An adapted, more extensive pre-interventional examination of these patients, including urine cultures and rectal swabs, seems to be very useful for patients undergoing TRBP, as also recommended in the PREVENT trial [2]. The choice of another antibiotic prophylaxis must also be considered in this group. Worldwide, and in European countries other than Germany, fosfomycin is approved for the indication of prophylaxis in TRBP and would therefore be the drug of choice. In Germany, fosfomycin is not approved for this indication, and the choice is, therefore, more difficult due to the lack of sufficient other alternatives. An alternative cannot be reliably named, but as mentioned above, in patients with diabetes, a rectal swab and urine culture must be performed before TRBP and the choice of antibiotic prophylaxis.

## 4. Materials and Methods

Prospective data collection including retrospective analysis was conducted. A total of 359 patients from the Department of Urology of the University of Regensburg, at Caritas-St. Josef Medical Center as a single center, included all men who underwent a TRBP in the period from 2 July 2019 to 30 June 2020 due to suspected carcinoma of the prostate.

The post-interventional hospitalization rate due to bacterial complications was relevant to our evaluation of antibiotic prophylaxis with amoxicillin/clavulanic acid. Hospitalization for bacterial complications after TRBP included febrile urinary tract infections (temperature ≥ 38 °C) and acute prostatitis with subsequent hospital treatment. The antibiotic was administered at a dose of 875 mg/125 mg every 8 h on the day before the biopsy, the day of the biopsy and the three days following TRBP.

The patient files served as the basis for the data collection, and data from anamnesis forms, doctors’ letters, examinations and intervention reports were taken from the patient administration program MCC^®^ (Meierhofer). Follow-up was performed for 30 days after TRBP.

Patient data were recorded using Excel^®^ Version 2408 (Microsoft, Redmond, WA, USA), and data analysis was performed using SPSS24^®^ (IBM, Armonk, NY, USA). Tests used were the *t*-test, Mann–Whitney U test, and Fisher’s exact test. The *t*-test is a statistical test that tests whether there is a significant difference between the means of two groups. The Mann–Whitney U test is used when the requirements for the *t*-test are not met. It tests whether the central tendencies of two independent samples are different. Fisher’s exact test is a nonparametric test for the examination of associations between two categorical, binary variables.

Patients with another prophylaxis, inappropriate application and/or dosage of prophylaxis and patients with incomplete data sets and who were not amendable to follow-up were excluded.

Immunosuppressive risk factors included leukemia, but also medical immunosuppression in autoimmune diseases, e.g., Rheumatism, and especially diabetes. The urogenital risk factors included any pathogenic diseases of the urinary tract, and the cardiovascular risk factors included coronary heart disease (CHD), peripheral arterial obstructive disease (PAOD), and inflammation or cardiac arrhythmias.

Amoxicillin/clavulanic acid was chosen after consultation with the Department of Microbiology and Hygiene of the University of Regensburg and the in-house pharmacy as a broad-spectrum beta-lactam-antibiotic in combination with a beta-lactamase-inhibitor with broad effectiveness against gram-negative bacteria. At the same time, the use of the substance in men and urinary tract infections was allowed.

## 5. Conclusions

The rate of hospitalization for infectious complications after TRBP under prophylaxis with amoxicillin/clavulanic acid was low in this retrospective single-center study.

The PREVENT trial showed that TRBP is still of relevance regarding infectious complications [2]. In times of obsolescence of ciprofloxacin and fosfomycin limitation in Germany, amoxicillin/clavulanic acid is an option with at least tendentially comparable complication rates compared to ciprofloxacin, but probably higher rates than fosfomycin [10,11,12,13,14,15,16].

Also, the use of the substance is practicable for patients, especially during their first appearance at a hospital for a biopsy before prophylaxis.

## Figures and Tables

**Table 1 antibiotics-13-00940-t001:** Patient characteristics.

Criteria	Total (*n* = 359)	No Complications (*n* = 349)	Complications (*n* = 10)	*p*–Value
Mean age (SD) [years]	66.9 (8.1)	67.0 (8.1)	65.4 (7.8)	0.870
Diabetes, *n* (%) Yes No	27 (7.5) 332 (92.5)	24 (6.9) 325 (93.1)	3 (30) 7 (70)	**0.032**
Immunosuppression, *n* (%) Yes No	3 (0.8) 356 (99.2)	3 (1) 346 (99)	0 (0) 10 (100)	0.919
Urogenital risk factors, *n* (%) Yes No	12 (3.3) 347 (96.7)	12 (3.4) 337 (96.6)	0 (0) 10 (100)	0.709
Cardiovascular risk factors, *n* (%) Yes No	147 (40.9) 212 (59.1)	141 (40.4) 208 (59.6)	6 (60) 4 (40)	0.179
Suspect DRE, *n* (%) Yes No	51 (14.2) 308 (85.8)	51 (14.6) 298 (85.4)	0 (0) 10 (100)	0.212
Median prostate volume (range) [cc]	48.0 (13–241)	47 (13–241)	67.5 (30–97)	0.142
Median PSA (range) [ng/mL]	7.8 (0.3–2351)	7.6 (0.3–2351)	9.1 (4,1–60)	0.332
Prior rectal swab, *n* (%) Yes No	8 (2.2) 351 (97.8)	7 (2) 342 (98)	1 (10) 9 (90)	0.204
Antibiotic therapy within the prior 6 months, *n* (%) Yes No	7 (1.9) 352 (98.1)	7 (2) 342 (98)	0 (0) 10 (100)	0.819
Urinary tract infections within the prior 12 months, *n* (%) Yes No	7 (1.9) 352 (98.1)	7 (2) 342 (98)	0 (0) 10 (100)	0.819
Prior biopsies, *n* (%) Yes No	112 (31.2) 247 (68.8)	108 (30.9) 241 (69.1)	4 (40) 6 (60)	0.382
Fusion biopsy, *n* (%) Yes No	303 (84.4) 56 (15.6)	293 (84) 56 (16)	10 (100) 0 (0)	0.179
Median number of taken cores (range)	14 (7–20)	14 (7–20)	14 (14–18)	0.446
PCA histology, *n* (%) Yes No	277 (77.2) 82 (22.8)	268 (76.8) 81 (23.2)	9 (90) 1 (10)	0.292
Prostatitis in histology, *n* (%) Yes No	165 (46) 194 (54)	163 (46.7) 186 (53.3)	2 (20) 8 (80)	0.086

DRE, digital rectal examination; PSA, prostate-specific antigen; PCA, prostate cancer; statistic measurements for *p*-values: *t*-test, Fisher’s exact test, Mann–Whitney U test; bold values statistically significant (*p* < 0.05).

## Data Availability

All data generated or analyzed during this study are included in this article. Further inquiries can be directed to the corresponding author.

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
