# Peer review of "Amoxicillin/Clavulanic Acid in Transrectal Biopsy of the Prostate—An Alternative in Times of Ciprofloxacin Obsolescence and Fosfomycin Limitation?"

_antibiotics, 2024, doi:10.3390/antibiotics13100940_

Round 1

Reviewer 1 Report

Comments and Suggestions for Authors

The authors aimed to test the amoxicillin/clavulanic acid  as an alternative option in antibiotic prophylaxis of trans-rectal biopsies (TRBP) within a single institution in Germany. The results relied on 359 patients. Of those, only ten (2.8%)  patients had an infection with hospitalization post-TRBP. However, the definition of complication has not provided and should be stated better. Moreover, they concluded that the current infection rate after TRBP was similar to the previous infection rates under ciprofloxacin. The conclusions were carefully reported due to the limitation of the study design. However, a sample size of 10 patients who exhibited complications, and even the absence of a multivariable model that adjusted for clinical variables (due to statical limitations) are two major concerns that cannot be ignored. How did the authors explain that? A discussion section may be implemented. Any data were available on the prostate gland asymmetry or on the difficult anatomy of the patient (PMID= 39122911). Moreover, TRBP time should also be a contributing factor to complications and infections as well as the expertise of biopsy operator. Any data on this regarding? These aspects should be ackowledged.

Author Response

The authors aimed to test the amoxicillin/clavulanic acid  as an alternative option in antibiotic prophylaxis of trans-rectal biopsies (TRBP) within a single institution in Germany. The results relied on 359 patients. Of those, only ten (2.8%)  patients had an infection with hospitalization post-TRBP.

However, the definition of complication has not provided and should be stated better.

Answer: We agree, please see the highlighted changes in the Material and Methods section in the lines 161-163.

Moreover, they concluded that the current infection rate after TRBP was similar to the previous infection rates under ciprofloxacin. The conclusions were carefully reported due to the limitation of the study design. However, a sample size of 10 patients who exhibited complications, and even the absence of a multivariable model that adjusted for clinical variables (due to statical limitations) are two major concerns that cannot be ignored. How did the authors explain that? A discussion section may be implemented.

We agree, please see the highlighted changes in the Discussion section in the lines 120-126 and int he Conclusions section line 189.

Any data were available on the prostate gland asymmetry or on the difficult anatomy of the patient (PMID= 39122911).

Answer: Unfortunately, there were no data available regarding this but only if DRE was suspect, please see the highlighted changes in the discussion section in the lines 129-131.

Moreover, TRBP time should also be a contributing factor to complications and infections as well as the expertise of biopsy operator. Any data on this regarding? These aspects should be ackowledged.

Answer: Unfortunately, there were no data available regarding this, please see the highlighted changes in the discussion section in the lines 127-128.

Reviewer 2 Report

Comments and Suggestions for Authors

Regarding manuscript "Antibiotic prophylaxis with amoxicillin/clavulanic acid in transrectal biopsy of the prostate – an alternative in times of ciprofloxacin- obsolescence and fosfomycin-limitation?" I have a few observations for the authors:

The manuscript requires major revision, both in the abstract and in the article. The idea of the study is good, but the description of the study needs improvement.

Both in the abstract and in the article, the purpose of the study and the design must be well mentioned.

In the Materials and methods are also put some of the results.

Where was the study conducted? Is it a single or multicenter study? The Materials and Methods will describe the study with all the variables recorded (see the variables in Table 1 of the Results).

The last paragraph from introduction (rows 50-54) can be moved in Materials and methods.

Table 1 it is not mentioned in the text.

Author Response

Regarding manuscript "Antibiotic prophylaxis with amoxicillin/clavulanic acid in transrectal biopsy of the prostate – an alternative in times of ciprofloxacin- obsolescence and fosfomycin-limitation?" I have a few observations for the authors:

The manuscript requires major revision, both in the abstract and in the article. The idea of the study is good, but the description of the study needs improvement.

Both in the abstract and in the article, the purpose of the study and the design must be well mentioned.

Answer: We agree, please see the highlighted changes in the abstract in line 16 and in the introduction in the lines 51-52

In the Materials and methods are also put some of the results.

Answer: Thank you, please see the highlighted changes in the manuscript in the lines 54-68.

Where was the study conducted? Is it a single or multicenter study?

Answer: University single center, please see the highlighted changes in the lines 16-17 in the abstract and 51-52 in the Introduction section, and in line 157 in Material and Methods section.

The Materials and Methods will describe the study with all the variables recorded (see the variables in Table 1 of the Results).

Answer: We agree, please see the changes in the Results section in the lines 89-95.

The last paragraph from introduction (rows 50-54) can be moved in Materials and methods.

Please see the higlighted changes in the manuscript in the Material and Methods section lines 179-183.

Table 1 it is not mentioned in the text.

Answer: thank you, see in the Results section line 96.

Round 2

Reviewer 1 Report

Comments and Suggestions for Authors

The authors addressed the concern raised before. Despite the limitation, it now achieves sufficient priority to be published. 

Comments on the Quality of English Language

Spelling errors and grammar mistakes were detected. I.e. Line 124 "analyzes"

Author Response

Spelling errors and grammar mistakes were detected. I.e. Line 124 "analyzes"

Thank you, please see the highlighted changes in line 124 and table 1.

Reviewer 2 Report

Comments and Suggestions for Authors

The authors have improved the content of the article, but the manuscript still requires some details:

First, the University Center/hospital where the study was conducted is not mentioned.

In the abstract, on lines 19-21, the sentences are similar "Ten (2.8%) of the 359 patients had an infection with hospitalization post TRBP. 349 (97.2%) patients had no infection-related hospitalization. Ten patients from a collective of 359 patients showed post-interventional complications. This corresponds to an incidence rate of only 2.8%." 

In the Introduction, the PREVENT study should be briefly described for easier understanding of the text.

In the Results, line 91, refers to urinary tract infections in the last 3 months before the TRBP, but in Table 1 there are data on UTIs in the last 12 months before the TRBP.

In line 108, "3MRGN" is just an abbreviation with no translation.

In Discussion, the paragraph "A high correlation could be observed in patients having diabetes as a comorbidity, as three patients with an infectious hospitalizing were diabetics (p=0.032)." (line 142-143) could be moved before the last paragraph (before line 150).

The statistical tests applied should be detailed in the Materials and Methods.

Could it be concluded that for people with diabetes another antibiotic should be chosen for the prevention of infections after TRBP or more extensive pre-interventional examination of these patients including urine-culture and rectal swabs?

Author Response

The authors have improved the content of the article, but the manuscript still requires some details:

First, the University Center/hospital where the study was conducted is not mentioned.

Answer: Please see the highlighted changes in lines 17-18 and 163-164

In the abstract, on lines 19-21, the sentences are similar "Ten (2.8%) of the 359 patients had an infection with hospitalization post TRBP. 349 (97.2%) patients had no infection-related hospitalization. Ten patients from a collective of 359 patients showed post-interventional complications. This corresponds to an incidence rate of only 2.8%." 

Answer: Please see the highlighted changes in lines 20-21

In the Introduction, the PREVENT study should be briefly described for easier understanding of the text.

Answer: Please see the highlighted changes in lines 35-37

In the Results, line 91, refers to urinary tract infections in the last 3 months before the TRBP, but in Table 1 there are data on UTIs in the last 12 months before the TRBP.

Answer: Please see the highlighted changes in line 93

In line 108, "3MRGN" is just an abbreviation with no translation.

Answer: Please see the highlighted changes in lines 110-111

In Discussion, the paragraph "A high correlation could be observed in patients having diabetes as a comorbidity, as three patients with an infectious hospitalizing were diabetics (p=0.032)." (line 142-143) could be moved before the last paragraph (before line 150).

Answer: Please see the highlighted changes in lines 151-152.

The statistical tests applied should be detailed in the Materials and Methods.

Answer: Please see the highlighted changes in lines 177-183

Could it be concluded that for people with diabetes another antibiotic should be chosen for the prevention of infections after TRBP or more extensive pre-interventional examination of these patients including urine-culture and rectal swabs?

Answer: Please see the highlighted changes in lines 157-158 and also the existing section 153-157.